# The Shape of Human Red Blood Cells Suspended in Autologous Plasma and Serum

**DOI:** 10.3390/cells11121941

**Published:** 2022-06-16

**Authors:** Thomas M. Fischer

**Affiliations:** 1Department of Experimental Physics, Saarland University, Campus E2 6, 66123 Saarbrücken, Germany; thmfischer@gmail.com; Tel.: +49-160-2293318; 2Laboratory for Red Cell Rheology, Krummer Weg 20, 52134 Herzogenrath, Germany

**Keywords:** thickness across the dimple, spontaneous curvature, aspect ratio, non-uniform rim thickness, shape normalization, reference configuration of the membrane skeleton, remodeling of the membrane skeleton

## Abstract

(1) Background: In almost all studies of the shape of the human red blood cell (RBC), the suspending medium was a salt solution supplemented with albumin. However, the ratio of thickness across the dimple region to the thickness of the rim (THR) depends on the albumin concentration. Values of the THR in the literature range from 0.27 to 0.627 whereas in the present work it was 0.550 or 0.601 whether measured in plasma or serum. (2) Methods: 9911 RBCs of eight donors were suspended in autologous plasma or serum. Sedimented RBCs were observed under bright field illumination at 416 nm. From the profiles of gray value, the THR was determined. (3) Results: The THR displays a wide distribution within a single blood sample. A direct correlation of THR and spontaneous curvature of the membrane is likely. The variation of the mean THR between different donors is large. The aspect ratio of RBCs viewed face-on ranged on average from 1 to 1.48. In oval RBCs, the rim is thicker along the major axis than along the minor axis, an effect increasing with increasing aspect ratio. Remodeling of the membrane skeleton occurs in vivo with a characteristic time (τ) on the order of 1 h. (4) Conclusions: Consideration of these data in models of RBC behavior might improve the agreement with observations. τ≈1 h suggests a more general type of reference configuration of the membrane skeleton than a stress free shape.

## 1. Introduction

About 50% of the human blood consists of red cells (RBCs). At rest, they assume the shape of a biconcave disk on average 8 µm in diameter and 2 µm thick. Their simple structure, a flaccid membrane bag filled with a concentrated hemoglobin (Hb) solution, allows RBCs to pass constrictions in the circulation on one hand and also to model RBC flow in silico on the other. Usually, these models use a standard RBC as a reference. In addition to the surface area and the volume, it is the depth of the concavities, called the dimples, on both sides of the disk that characterizes such a standard shape. An obvious parameter to quantify this depth is the ratio of the thickness across the dimple region to the thickness of the rim. This ratio is referred to as thickness ratio (THR) hereafter.

The standard RBC used almost exclusively in modeling is the experimental shape of Evans and Fung [1]. Its THR is 0.314. This study was about a demonstration of the method and covered only a single blood donor. As a side note, it is mentioned here that the data of the much more extensive study based on the same method [2] were not used. Having studied RBCs for decades, I was never content with the choice of the Evans–Fung-shape because its THR was much lower than what I saw under the microscope.

However, what could be the origin of that discrepancy in consideration of the superior method of measurement of Fung and coworkers? I suggest the step that precedes the measurement proper, i.e., the method to prepare the RBC sample. In the studies of Fung and coworkers [1,2] as in most other studies of RBC shape, the cells were suspended in buffered salt solutions supplemented with a small amount of albumin to prevent crenation due to the so-called glass or surface effect [3]. However, the problem with this kind of suspending phase is that the RBC shape in particular its THR depend on the concentration of albumin. The higher the concentration of albumin the lower the THR.

This is demonstrated in a movie (Appendix A) where the albumin concentration was ramped up in a continuous fashion by a diffusive process. Details of the experimental protocol are given in Appendix A. Figure 1 shows selected frames from that movie. The shape changes from echinocyte stage 2 [4] via biconcave shapes to stomatocyte stage 2 [4]. From 0 s to 81 s, the numerous spikes are converted to a few wide bumps. From 107 s to 343 s, the gross shape is biconcave with the THR continuously decreasing.

By using autologous plasma and serum as suspending media, the present study avoids the uncertainty as to which albumin concentration results in the natural value for the THR. With plasma or serum there is no glass effect. Unavoidable aggregates were eliminated during processing of the microscopic images. The initial goal was to determine the THR. It is found that the average THR in plasma is 0.550 and as such clearly larger than the value 0.314 used up to date, i.e., the RBCs are flatter. The microscopical method as well as the large number of RBCs studied allow to make statements that to the best of my knowledge have not been published before.
A direct correlation of THR and spontaneous curvature of the membrane is likely.The variation of the mean THR between different donors is large.The aspect ratio of RBCs viewed face-on ranged on average from 1 to 1.48.In oval RBCs, the rim is thicker along the major axis than along the minor axis, an effect increasing with increasing aspect ratio.Remodeling of the membrane skeleton occurs in vivo with a characteristic time on the order of 1 h.

## 2. Materials and Methods

### 2.1. Blood Sampling and Preparation

The use of human blood was in accordance with the Code of Ethics of the World Medical Association (Declaration of Helsinki). Venous blood was drawn from the antecubital vein of healthy donors. During blood collection, the plunger was withdrawn slowly in order to avoid hemolysis which might occur upon quick withdrawal when the jet of blood hits the plunger. The age of the donors ranged from 25 to 75, five males and three females. Blood was collected before lunch in order to obtain clear plasma. Two types of vacutainers were used; one sodium-heparinized (S-Monovette 9ml LH, Sarstedt, REF 02.1065) to obtain plasma and RBCs. The other vacutainer was without anticoagulant (S-Monovette 9 mL Z, Sarstedt, REF 02.1063.001) for the preparation of serum. After collection, the serum vacutainer was stored upright at room temperature. Three preparations were studied and identified by the codes p, s, and sw.
codep Heparinized blood was pipetted into an Eppendorf vial and centrifuged at 5700 g for 8 min. The clear plasma supernatand was transferred into another Eppendorf vial. To obtain the final concentration of RBCs, heparinized blood was diluted with plasma in three steps 500 to 1000-fold.codes After 30 min at room temperature, the serum vacutainer was centrifuged in a swing-out-rotor at 2000 g for 10 min. The clear serum supernatand was transferred into an Eppendorf vial. Heparinized blood was diluted with serum in the same way as described above for plasma.codesw Heparinized blood was washed three times with phosphate buffered saline (PBS, pH 7.4 1×, gibco, life technologies). A 50% suspension of the washed RBCs in PBS was diluted with serum in the same way as described above for plasma.

The experiments were done at room temperature and completed 2–3 h after blood collection.

### 2.2. Microscopic Observation

A chamber for microscopic observation was prepared with a slide, a coverslip, and parafilm as spacer on opposite sides of the coverslip. The gap between slide and coverslip was approximately 100 µm. The RBC suspension was applied with a pipette on one of the open sides of the chamber thus entering the gap readily due to capillary action. Immediately after filling, the chamber was inverted to prevent sticking of RBCs to the surface of the slide. After 2.5 min, the chamber was inverted again and its two open sides were sealed with silicone grease (Korasilon-Paste niedrigviskos, Kurt Obermeier, Bad-Berleburg, Germany) to prevent evaporation. After another inversion, it was placed on an x,y-table of an inverted microscope (Leitz, Diavert, Wetzlar, Germany). The microscope was equipped with an immersion lens (100/1.3 NPl) and an immersion condenser (1.25 NA). For bright-field illumination, a high power LED (OCU-440 UE410, OSA Opto Light, Berlin, Germany) was used. Its peak wavelength of 416 nm covered the soret band where the absorption of Hb is maximal. As collector served a lens with 17 mm focal length allowing to light the back focal plane of the condenser completely. Moving the stage by hand, images were taken with a b/w camera (DMK 33UX252, The Imagingsource, Bremen, Germany) and stored in the hard disk of a computer.

### 2.3. Image Processing

Image processing was done with Imagej [5]. For a detailed description see Appendix A. The essential steps are demonstrated here by means of an RBC with average values of aspect ratio and THR. Figure 2a is an image cropped from the recorded frame. In the center is the RBC in question. In the upper left hand corner of Figure 2a, part of a neighboring RBC is visible. The function of the red circles is explained below. To arrive at the image in Figure 2b, the RBC in the center is rotated to make the long axis horizontal. The rotated image is binarized using a threshold in gray value (GV). The dimensions of a rectangle (not shown) enclosing the binarized object in the center of Figure 2b are taken as the major and minor axes of the RBC. Figure 2c shows the original image after normalizing the GVs in such a way that the background is white. This procedure used the GVs inside the red circles in Figure 2a as the original background values. Sections along major and minor axes in Figure 2c resulted in profiles of GV. These profiles were obtained from the GVs inside slender rectangles shown for the major axis in magenta (Figure 2c). The GVs are shown in Figure 3 as blue circles. The red line is a spline through these data. Three GVs are extracted from the splines along the major and minor axis (long and wide in Figure 3), respectively. The two minima correspond to the GVs of the rim. The maximum in between corresponds to the GV of the dimple. The quantities calculated from these six values are described in Section 2.5.

**Figure 3 cells-11-01941-f003:**
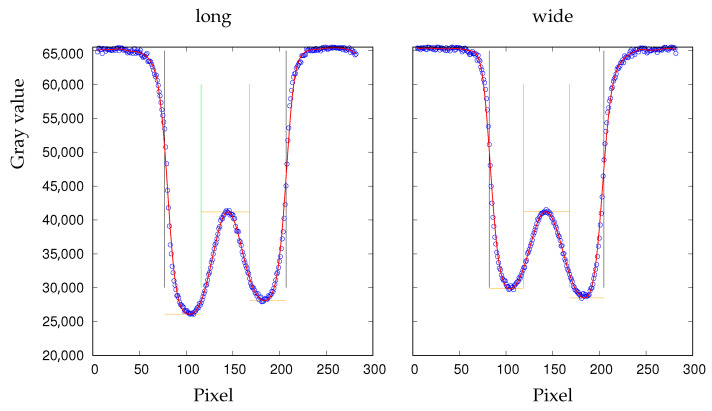
Sections along the major (long) and minor (wide) axis through an RBC with median values (Table 1) of THR and aspect ratio. The vertical black lines indicate the edges of the digitized shape (Figure 2b). The red line is a spline through the GVs (blue circles). The vertical green lines are auxiliary lines for the determination of the local maximum of the red line and the two minima at its left and right hand side. These three values are indicated by horizontal orange lines.

The shape of the curves in Figure 3 may be to counter-intuitive. Contrary to a cross section through a real red cell the curves are inverted. The reason being, darker pixels have lower GVs.

### 2.4. Sorting Poikilocytes

Non-biconcave RBCs summarized here as poikilocytes were excluded from the evaluation of mean values by appropriate filters. Five types of poikilocytes were distinguished.
Stomatocytescan be described as cup-shaped RBCs (cf. Figure 1 last image). Their profile of GVs shows two minima and a maximum just as the profile of biconcave discocytes. However, what appears to be the rim or dimple in the profiles is actually the wall or bottom of the cup, respectively. The profile is less curved in the “dimple” region and the transition from the “rim” to the “dimple” is abrupt unlike in discocytes. Examples of the filtering procedure are shown in Appendix A. A typical stomatocyte is shown in Figure 4a.Echinocytes type 1 (E1)oriented edge-on and sectioned microscopically through the center appear biconcave. In the face-on orientation, their outline is wavy. An example of the filtering procedure is shown in Appendix A. A typical E1 is shown in Figure 4b.Echinocytes type 2 (E2)are similar to E1 in that their overall shape is flat and their outline is wavy in the face-on orientation. In addition, E2s show spicules (cf. Figure 1 first image) or bumps (cf. Figure 1 second image) extending away from the mid-plane. In the present experiments these bumps are recognized as a local increase in GV. An example of the filtering procedure is shown in Appendix A. A typical E2 is shown in Figure 4c.Pitting type 1 (P1):The outline of some of these RBCs presented a single indentation in the face-on orientation. In others the curvature of the outline was zero or negative in places (the outline of a circle is defined as positive).The nomenclature (pitting) relates to the suggested origin of these shape (Section 4.7.2 and Section 4.7.3). A prominent example is shown in Figure 4d and all P1s found are shown in Appendix A.Pitting type 2 (P2):Some of these RBCs showed a tongue protruding from the body of the RBC. Others were unusually small. A prominent example is shown in Figure 4e and all P2s found are shown in Appendix A.

Many of the P1s and P2s would not have passed the filters applied to all RBCs and described in Appendix A; e.g., are out of focus or even aggregated. They were saved because they appeared to be of interest with respect to the way the spleen removes RBCs from the circulation. Of all poikilocytes, only E1 allowed to calculate a meaningful value of the THR.

### 2.5. Calculations

The basis for converting GV to RBC thickness is the Beer–Lambert law:(1)$$I_{1}=I_{0}e^{-\mu d},$$
where $I_{0}$ is the incident and $I_{1}$ the transmitted intensity of light, μ the absorption coefficient, and *d* the RBC thickness. The ratio $I_{1}/I_{0}$ is the transmittance.

The linearity between intensity of illumination on the camera chip and GV in the images is demonstrated in Appendix A. Thus, the GV of the background (65,535 after normalization of the GVs) could be used as $I_{0}$ and the GVs of the RBCs as $I_{1}$. Since the Hb concentration of the RBCs was not known, values for the thickness of the RBCs could not be deduced. Thickness *ratios* within the same RBC, on the other hand, were calculated according to:(2)$$\frac{d_{a}}{d_{b}}=\frac{\ln\left({\mathrm{GV}}_{a}\right)-\ln\left(65535\right)}{\ln\left({\mathrm{GV}}_{b}\right)-\ln\left(65535\right)},$$
where $d_{a}$ and $d_{b}$ indicate two values for RBC thickness. Besides the RBC, the incident light passes through about 100 µm plasma. Accounting for the absorption in plasma, however, does not alter Equation (Equation 2) as detailed in Appendix A. Using Equation (Equation 2) and the simplified notation in Figure 5, the following quantities were calculated:
(3)$${\mathrm{THR}}_{\mathrm{long}}=\frac{2d_{\ell }}{\ell_{1}+\ell_{2}},$$
(4)$${\mathrm{THR}}_{\mathrm{wide}}=\frac{2d_{w}}{w_{1}+w_{2}},$$
(5)$$\mathrm{THR}=\frac{2(d_{\ell }+d_{w})}{\ell_{1}+\ell_{2}+w_{1}+w_{2}},$$
(6)$$\frac{{\mathrm{THR}}_{\mathrm{long}}}{{\mathrm{THR}}_{\mathrm{wide}}}=\mathrm{LFA}\approx \frac{w_{1}+w_{2}}{\ell_{1}+\ell_{2}},$$
where the abbreviation LFA is introduced in Section 3.2. The sign ≈ appears because $d_{\ell }$ and $d_{w}$ are slightly different due to inevitable fluctuations in RBC thickness (Figure 6c and Appendix A) and the spline through the measured data.
(7)$$\mathrm{Ratio}\;\mathrm{between}\;\mathrm{rim}\;\mathrm{thicknesses}\;\mathrm{at}\;\mathrm{the}\;\mathrm{two}\;\mathrm{extremities}\;\mathrm{of}\;\mathrm{the}\;\mathrm{major}\;\mathrm{axis}\;=\;\frac{\ell_{1}}{\ell_{2}},$$
where $\ell_{1}<\ell_{2}$.
(8)$$\mathrm{Ratio}\;\mathrm{between}\;\mathrm{rim}\;\mathrm{thicknesses}\;\mathrm{at}\;\mathrm{the}\;\mathrm{two}\;\mathrm{extremities}\;\mathrm{of}\;\mathrm{the}\;\mathrm{minor}\;\mathrm{axis}\;=\;\frac{w_{1}}{w_{2}},$$
where $w_{1}<w_{2}$.

## 3. Results

### 3.1. Distributions and Correlations

9911 RBCs of 8 donors entered the study. 4882 were suspended in plasma and 5029 in serum. For each donor and suspending medium, distributions of the following six quantities
the projected area;the aspect ratio;the THR;the mean transmittance;the ratio between rim thicknesses at the two extremities of the major axis;the ratio between rim thicknesses at the two extremities of the minor axis;

are displayed in data sheets. An example of the first page is shown in Figure 7. Please note the logarithmic scale on the ordinate of the distributions. The value of 0.6 on the ordinate serves as a pseudo zero. The shaded gray histograms represent the E1s found for the respective donor. They did not enter the calculation of the mean values. For the sake of brevity, “Rim Thickness Ratio along the major axis” is chosen as axis label instead of “ratio between rim thicknesses at the two extremities of the major axis”. The analogue applies to the minor axis.

The distribution of aspect ratios extended in some rare cases to values < 1 (cf. Figure 7). The origin of this illogicality is due to image processing. The orientation of the major axis is determined from an ellipse fitted to the contour whereas the lengths of the axes and the aspect ratios are determined from the dimensions of a rectangle enclosing the binarized contour. Due to the discrete definition of the shape (in pixels), it can happen that the major axis of the ellipse becomes the shorter edge of the rectangle. Of course, it happens only if the aspect ratio is almost unity.

The following pages of the data sheets show the pairwise interdependence and correlation between these six quantities plus ${\mathrm{THR}}_{\mathrm{long}}/{\mathrm{THR}}_{\mathrm{wide}}$ named LFA (nomenclature explained in Section 3.2). This results in 21 scatter plots. The background of the 22nd scatter plot $|\delta {\mathrm{focus}}_{\mathrm{wide}}|/|\delta {\mathrm{focus}}_{\mathrm{long}}|$ versus Aspect Ratio is detailed in Appendix A. Finally, images of the P1s and P2s found in the sample of the respective donor are displayed.

The bundled data sheets of all donors are shown in a PDF (Appendix A). As a cross-sectional reference, the distributions of Projected Area for all donors are bundled in Appendix A and likewise the distributions of Aspect Ratio, THR, and Mean Transmittance in Appendix A.

In each scatter plot, the respective correlation coefficient is shown in orange in the lower left corner. The ranges of the correlation coefficients of the 21 scatter plots of all donors (plasma and serum lumped together) are graphically displayed in Appendix A. Selected correlations are discussed below.

### 3.2. Lady’s Finger Asymmetry

As an example, the scatter plot ${\mathrm{THR}}_{\mathrm{long}}/{\mathrm{THR}}_{\mathrm{wide}}$ versus Aspect Ratio of donor 4 for plasma suspension is shown in Figure 8. The red line is a regression of all data with aspect ratios inside the interval 〈median − 0.04, median + 0.04〉, where the value of 0.04 was chosen pragmatically common for all plots of this type and the median is taken from Figure 7. The red cross indicates the middle of the regression line, i.e., the median of Aspect Ratio.

${\mathrm{THR}}_{\mathrm{long}}/{\mathrm{THR}}_{\mathrm{wide}}$ is essentially the ratio of rim thicknesses wide/long. Values of this ratio <1 indicate that the rim is thicker along the major axis than along the minor axis. The respective 3D shape with such an asymmetry is reminiscent of a cookie called Lady’s Finger. The value of ${\mathrm{THR}}_{\mathrm{long}}/{\mathrm{THR}}_{\mathrm{wide}}$ is referred to as Lady’s Finger Asymmetry (LFA) hereafter.

The y-position of the red cross in Figure 8 denotes the value of LFA corresponding to the median of Aspect Ratio. Its value is shown in red in the lower right corner of each plot of this type, where <> indicate the correspondence to the median of Aspect Ratio (cf. Figure 8).

To illustrate observed aspect ratios and values for <LFA>, five RBCs are shown in Figure 9 located equally spaced on the regression line in Figure 8 and its extension to smaller and larger values of Aspect Ratio.

### 3.3. Mean Values

RBCs suspended in plasma were studied for all eight donors. For serum, two different protocols were applied: code s for donors 1 and 3 and code sw for donors 3, 4, 5, 6, 7, and 8. Before proceeding, the two serum values of donor 3 were averaged.

Table 2 shows the mean values of the means, the mean values of the standard deviations (SDs), and the mean values of the coefficients of variation (CVs) determined for each distribution.

Table 1 shows the mean values, SDs, and CVs of the medians determined for each distribution. Here, the SDs are much smaller than in Table 2. By analogy to the standard error of the means, the SDs could be called the standard error of the medians.

The mean value, SD, and CV of <LFA> are shown in Table 3. Although <LFA> is not a true mean value but is read from from a regression line, the SD in essence is a standard error of the means. A statistical evaluation of the difference between plasma and serum as suspending media is performed on the basis of the individual donors in Section 4.5.

The distributions of the quantities in Table 1, Table 2 and Table 3 are shown in Appendix A, respectively. As expected, means and medians differ appreciably for asymmetric distributions.

## 4. Discussion

The resting shape of the human RBC was the subject of numerous experimental studies in the past. In almost all of the previous studies, RBCs were suspended in PBS supplemented with albumin. Notable exceptions are studies using serum as suspending medium [6,7,8]. As shown in the Introduction, PBS plus albumin can lead to a systematic change of the ratio between the thickness across the dimples to the thickness of the rim (THR) compared to its natural value. Even the use of serum is not free of such a bias as shown below. Except for early measurements of RBC diameter [6,9], the present study is to the best of my knowledge the first to measure RBC shape in its natural habitat the blood plasma.

### 4.1. Method

The present method uses the absorption of Hb in the soret band. The outline of RBCs oriented face-on was found as the border inside which the absorption was above a threshold. The size of the area inside this border depends on the type of the threshold criterion. This dependence, however, has negligible influence on the deduced quantities: the qualitative shape and the aspect ratio. In addition shows the comparison with literature values of RBC diameter (Section 4.2) no bias in either direction.

The profile along the major and minor axes was determined from the course of the GVs. Due to the numerical aperture used in microscopical imaging, a GV at a certain location depends on the absorption in a double cone with an angle of 60°. Generally, this leads to a smoothing of the GV distribution depending on the local cell thickness and geometry. This disadvantage versus more sophisticated methods is outweighed by the fact that the present method does not require to fit the primary data to a theoretical shape which up to now always was circular symmetric [1,2,10,11,12].

### 4.2. RBC Diameter and Area

Previous studies reported a single value for the RBC diameter. In the present study major and minor axes of RBCs were measured. For the sake of comparison with the early results, these data were converted to a single RBC diameter. Two methods were used (i) $\langle d_{1}\rangle =\sqrt{major\;axis\cdot minor\;axis}$ (ii) $\langle d_{2}\rangle =\left(major\;axis+minor\;axis\right)/2$. Both values were close and therefore averaged.

Table 4 shows the present and earlier results. The neglect of aspect ratios >1 had different reasons. For cells evaluated in the edge-on orientation a single cross section is available only [7,13]. In the face-on orientation three strategies were pursued to obtain a single value for the RBC radius, (i) cells which were obviously not circular were not evaluated [9], (ii) minimal and maximal diameters were averaged [13], (iii) the numerical fitting suppressed deviations from circular symmetry [1,2,12].

The large range of the values for the average RBC diameter in Table 4 is attributed to the different methods used. The average RBC diameter is not a prime result of the present study. Nevertheless is the method to determine the threshold value during image processing approved by the fact that the calculated value of the average RBC diameter is within the range of the published values.

The mean CV of Projected Area among the eight donors is 0.1043 for plasma suspensions and 0.1049 for serum (Table 2). These values correspond well with the average of two donors for the CV of total surface area of 0.109 [14].

### 4.3. Influence of the External Environment on the Shape of the Red Cell

It has long been known that the RBC shape can be altered by various substances added to isotonic saline as suspending medium [15]. Of particular interest here are shape changes occurring close to artificial surfaces described already by Ponder [16] and traditionally called glass effect. The phenomenon was studied in detail by Eriksson [3]. The salient features are: (i) RBCs suspended in PBS or saline and sedimented on a glass-surface are transformed into echinocytes. (ii) With sparsely distributed RBCs, the effect is maximal, and with a complete coverage of the surface, almost not existent. (iii) With increase of the distance from the surface the extent of glass effect decreases. (iv) The shape change is reversible upon removal of the RBCs from the surface. (v) The glass effect can be elicited with a multitude of artificial surfaces. (vi) The glass effect can be prevented by 0.013% (*w*/*v*) albumin. In higher concentrations albumin acts stomatocytic.

### 4.4. Thickness Ratio (THR)

The mean values found in the literature are shown in Table 5. Two origins are suggested for their large range: (i) particularly at low numbers of studied RBCs, it could be the selection of nicely looking cells. (ii) as detailed in the Introduction and in Section 4.3, it could be the local environment. Considering the variation in Table 5 the most likely origin is (ii)—with one exception.

The low value reported in the first line compared to that in the last line is unexpected since the same suspending medium was used. The low value in the first line may be due to a false determination of the thickness across the dimple. In RBCs oriented edge-on, the apparent position of the membrane in the dimple region is biased by diffraction at the curved interface between cell interior and outside [17]. Even Ponder himself, the author of the study, used different tracking rules of the outline in his numerous studies on RBC dimensions [17]. In contrast to Ponder, the present determination of the THR used the distribution of GVs of RBCs oriented face-on.

The range of the observed values of THR within a blood sample is wide which is in keeping with earlier observations [13]. The average range of eight donors was from 0.16 to 0.89. The distributions of medians and means (Appendix A) show no systematic difference. The average of the respective mean values amounts to THR = 0.55 with SD = 0.06. The value is much larger than 0.314 [1] used traditionally in modeling RBC mechanics in silico.

The value of the THR is an important parameter in modeling because it is the result of a balance between two prime quantities in RBC mechanics: (i) The spontaneous curvature of the phospholipid bilayer a structure conferring bending elasticity to the membrane. The spontaneous curvature can be considered as the curvature a piece of membrane would assume when conceptually excised from the membrane (for more details cf. [18]). (ii) The reference configuration of the membrane skeleton a filamentous network of proteins laminated to the inner side of the bilayer which confers shear elasticity to the membrane. This reference configuration is usually modeled as stress-free shape.

A decrease in spontaneous curvature decreases the THR as does a stress-free shape approaching the sphere and vice versa [19,20]. Both quantities are unknown. The knowledge of the natural value of THR allows to limit their ranges.

The present results do not allow a straightforward statement as to which one of the two parameters is responsible for the observed variation in THR. However, comparison of the distributions of discocytes and E1s gives an indirect clue. In Appendix A and in Figure 7, it can be appreciated that in the histograms of the THR the E1s are consistently on the right hand side of the median value of the respective discocytes. Figure 10 shows the THR of E1s relative to the median of the respective discocytes summarizing all donors. On the other hand, it is known from modeling that the spontaneous curvature is higher in E1 than in discocytes [19,21,22]. Therefore, the preference of the E1s for large values of THR suggests a direct correlation of THR and spontaneous curvature.

Considering the distribution amongst the eight donors, the value of the CV of means and medians of THR amounts to 0.114 (cf. Appendix A). The value indicates an appreciable variation of mean THR between different donors. Corresponding to the direct correlation suggested above, this would apply to the mean spontaneous curvature as well.

### 4.5. Comparison between Plasma and Serum

For each donor, the medians obtained for serum suspension were divided by the respective plasma values. The distributions of these ratios are shown in Appendix A.

The hypothesis that the mean value of these distributions was equal to unity was tested with a two-sided one-sample t-test. On a significance level of 0.05, the hypothesis was accepted for all quantities except for the THR. This indicates that the spontaneous curvature of the RBC membrane is significantly greater in serum compared to the value in plasma. Since the value of the THR is a prime result of this study, only data of RBCs suspended in plasma are considered in the rest of the Discussion.

### 4.6. Correlations

Most correlation coefficients are are close to zero (Appendix A) and of minor interest. The absence of a correlation between THR and Aspect Ratio is in keeping with a previous result [23]. The inverse correlation of Mean Transmittance with THR is not surprising. It is expected that RBCs with deep dimples absorb less light. The mean correlation coefficient of −0.186 between THR and Projected Area indicates an insignificant trend towards shallow dimples at small projected areas. This trend is in keeping with the adaptation of RBC volume to the requirements of negotiating the restricted parts of the microcirculation secondary to the physiological decrease in surface area [24,25], i.e., the lower the surface area the more spherical shapes are tolerated.

### 4.7. Deviations from Circular Symmetry

#### 4.7.1. Determinants of RBC Shape

Besides the extrinsic quantities volume and surface area, the resting shape of the RBC is determined by an equilibrium of bending and shear elasticity of the membrane. Four quantities contribute to this equilibrium: two elastic moduli and two reference configurations. For bending the reference configuration is a spontaneous curvature considered to be constant on the membrane. For shear it could be a shape in which shear tensions vanish—the so-called stress free shape.

#### 4.7.2. P1s

Some of the RBCs classified as P1 show a bite morphology (Appendix A). Such RBCs have been observed after oxidative damage and designated as “bite cells” [26,27,28]. Their shape was interpreted as a consequence of the removal of Heinz bodies in the spleen. By analogy, the bite cells amongst the P1s are interpreted likewise.

Contrary to the previous notion [29], Ciana et al. [30,31] showed that in parallel with the loss of bilayer the respective amount of skeletal material is lost in the course of the decrease of surface area during the mature life of RBCs. The consequences for the membrane in the mother cell after removal of a patch forming the later vesicle are envisaged as follows. Due to its 2D liquid property the bilayer reseals with little or no change of its spontaneous curvature. Likewise, it can be expected that the membrane skeleton in the mother cell reseals by rebinding of open ends left within the skeletal network. The exotic shape of bite-cells suggests that the new reference configuration of the membrane skeleton owns essential features of the new shape.

The low frequency of such shapes suggests that they do not persist but return towards a normal outline with time. Besides bite cells, RBCs with a bean-like morphology or with partially straight outlines were classified as P1s as well. These shapes are interpreted as intermediates of the return to an outline with positive curvature. For such a normalization of the shape the reference configuration of the skeleton has to change. On a molecular level such a change requires that bonds within the network of proteins open and rebind in a new configuration. It seems reasonable to suppose that the frequency of opening increases with the stresses in the skeleton.

The frequency of P1s allows an order of magnitude estimate of the characteristic time (τ) of the envisaged shape normalization. The decrease of average surface area of 159.9 µm^2^ at the begin to 134.7 µm^2^ at the end of the mature RBC life [14] would require the loss of 8 or 32 spherical vesicles with diameters of 1 µm or 0.5 µm, respectively. In about 9900 RBCs, 43 RBCs with a bite morphology and intermediates towards return were observed. Assuming a bite every 15th day,
(9)$$\frac{\tau }{15\;d}=\frac{43}{9900}$$
results in τ≈ 1.6 h. On the other hand, inducing a strong deformation of resting cells in vitro resulted in τ⪆10 h [32]. In these experiments, stresses in the membrane skeleton originated in the mechanical equilibrium between bending and shear elasticity. The large difference suggests (i) stresses in resting cells are too low to initiate bond opening and (ii) a base level of spontaneous bond opening is virtually absent. Bond opening in vivo is therefore ascribed to deformations the RBCs are subjected to during tank-treading in arterioles or during squeezing through narrows in the circulation. A circulation time of 1 min results in about 100 passages of the microcirculation during τ and is in keeping with the hypothesis. Set free from the external constraints, the open bonds can rebind in a reference configuration conforming better to the bending elasticity of the bilayer. Of course, τ≈ 1.6 h does not invalidate the elastic behavior found in experiments of short duration [33].

#### 4.7.3. P2s

Some of these RBCs showed a tongue extending from the main cell body (Appendix A). The necessary remodeling of the membrane skeleton to achieve this shape is ascribed to large stresses when part of the RBC was held back in inter-endothelial slits of the spleen and the rest of the cell was pulled away by fluid drag. Similar tongues were observed in some of the bite cells observed after oxidative damage [26,27,28].

#### 4.7.4. Aspect Ratio

The range of the observed values of aspect ratio within a blood sample is wide. The average range of eight donors was from 1 to 1.48. Considering theoretical results on vesicle shapes, the origin of elongated RBC shapes as well as the LFA may be due to specific values of the spontaneous curvature of the bilayer. Amongst the shapes a pure bilayer would have are elongated shapes called prolates or pears depending on the spontaneous curvature [34]. Based on the mechanism suggested above, RBCs with such spontaneous curvatures could bring about a membrane skeleton with an elongated shape and in the case of pears with an LFA.

Playing devil’s advocate, an alternate explanation could be presented to explain oval shapes. One could suspect that the elongated appearance is the result of a slanted orientation of circular RBCs. Due to the shallow depth of field, this would become apparent by a difference (δ) of the focus of the outline at diametrical positions along the minor axis. As an appropriate parameter $|\delta {\mathrm{focus}}_{\mathrm{wide}}|/|\delta {\mathrm{focus}}_{\mathrm{long}}|$ was chosen. Its determination is described in Appendix A. The correlation coefficients of this parameter with Aspect Ratio range between −0.065 and 0.059 thus invalidating this suspicion.

### 4.8. Measurement

#### 4.8.1. Fluctuations of GVs

Two sources can produce fluctuations of the observed GVs: (i) thermal noise of the camera chip and (ii) thickness fluctuations of the RBCs, i.e., flickering. Both sources were evaluated by taking a movie with 400 frames/s of an RBC suspension in plasma prepared in the usual way. A clipping of a single red cell was evaluated in the same way as all red cells in this study.

Figure 6a shows the trace of the average GV of a square selection in the background with a side length of 4 pixel corresponding to the width of many rectangles used to obtain the GV profile. On top of these fluctuations add the thickness fluctuations. Figure 6b shows the trace of the GV of the average of the four rim positions. Similarly shows Figure 6c the time dependence of the average of the two dimple values. Finally, Figure 6d shows the THR.

Accounting for the different scale in Figure 6a compared to Figure 6b,c, it is obvious that the fluctuations due to noise of the camera chip can be neglected compared to those due to flickering. Comparison of Figure 6b,c shows that the thickness fluctuations in the dimple region are much larger than those of the rim. Please note that the scale of the ordinate in both panels is the same. The mirror symmetric course of the THR and the GVs of the dimple (Figure 6c,d) shows that the fluctuations of the THR are dominated by the thickness fluctuations of the dimple. The large fluctuation of the dimple thickness observed in this example is not in keeping with flicker measurements [35,36,37]. A search for the origin of this discrepancy is beyond the scope of the present study. For the sake of completeness, shows Appendix A the time series of GVs corresponding to the six locations indicated in Figure 5. Averages of these traces are shown in Figure 6b,c.

Figure 11a shows the angular diffusion. Figure 11c,d show the ratio between rim thicknesses at the two opposite ends of the major and minor axis, respectively. It is interesting to note that a ratio close to one can cross the line of unity (Figure 11c). This behavior does not show up in the respective distributions (cf. Figure 7) because there the smaller thickness was divided by the larger one whereas here it is the ratio of two individual thicknesses. In contrast to these variations, the aspect ratio is a constant attribute of RBC shape (Figure 11b). Despite some variation, this applies as well to the LFA (Figure 11e). The large fluctuation of the THR (Figure 6d) as well as the smaller ratios between rim thicknesses (Figure 11c,d) indicate that in unfixed RBC preparations only the mean value of a large number of RBCs is a meaningful result.

#### 4.8.2. Attachment to the Bottom of the Chamber

An attachment of the sedimented RBCs to the coverslip forming the bottom of the observation chamber would change their shape and to some extent the derived quantity the THR. However, plasma as the suspending medium prevented a firm attachment. This is demonstrated in Appendix A by the lateral diffusion of three RBCs in the observation chamber (Appendix A).

#### 4.8.3. Effect of Gravity

Due to the difference in density between cytoplasm and plasma [38], there is a small effect of gravity on the shape of the RBCs. Application of a theoretical result on phospholipid vesicles [39] however, shows that this effect can be neglected after inserting RBC specific quantities in the respective equation (Appendix A).

## 5. Conclusions

Implementation of the following points in future in silico models of RBC behavior might improve the agreement with observations.
Based on the averages of eight donors and plasma suspension, the following ratios are suggested for a mean resting RBC: aspect ratio 1.07, mean THR 0.55, <LFA> 0.942. This data results in a THR along the major axis of 0.5664 and along the minor axis of 0.5336. Based on an elliptical outline and the average projected area of 51.5 µm², the average major and minor axes amount to 8.38 µm and 7.83 µm respectively.To provide an analytical description of the average shape is beyond the scope of this report. For models with a discrete description of the membrane, a sequence of steps to design the average shape is presented in Appendix A. The procedure follows the approach of Canham [40], Helfrich [41], Deuling and Helfrich [42].Besides the distribution of the shear modulus of the membrane skeleton [43], the variation of the shape within the RBC population could be considered.Considering the short characteristic time of remodeling and the various shapes the RBCs assume during circulation, a more general concept for the reference configuration of the membrane skeleton could be adopted than that of a stress free shape.

## Figures and Tables

**Figure 1 cells-11-01941-f001:**
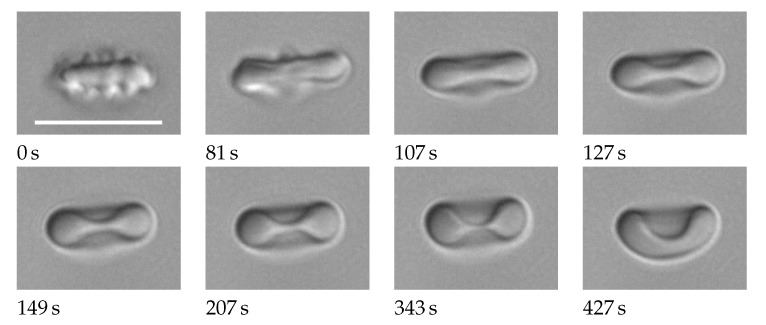
The shape of a red cell changes from echinocyte to stomatocyte. Photo montage from Appendix A. The suspending medium is phosphate buffered saline with a slowly increasing concentration of albumin. Below the images, the actual times of the process are given. The actual concentrations of albumin are not known. They are >0 at 0 s, <2 g/dl at 427 s, and monotonously increasing in between. The scale bar indicates 10 µm.

**Figure 2 cells-11-01941-f002:**
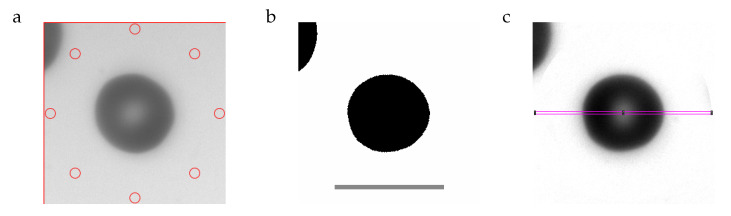
(**a**): cropped image with eight circular regions (red) used for normalizing the gray values (GVs). (**b**): digitized image. The scale bar indicates 10 µm. (**c**): image after normalizing the GVs. The rectangle (magenta) is used for evaluating the GV-profile. The RBC in (**b**,**c**) is rotated to make the major axis horizontal. For more details see text.

**Figure 4 cells-11-01941-f004:**
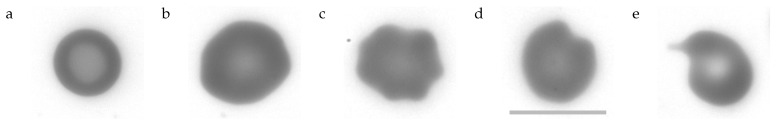
Typical shapes. (**a**): stomatocyte. (**b**): E1. (**c**): E2. (**d**): P1. (**e**): P2. The scale bar indicates 10 µm.

**Figure 5 cells-11-01941-f005:**
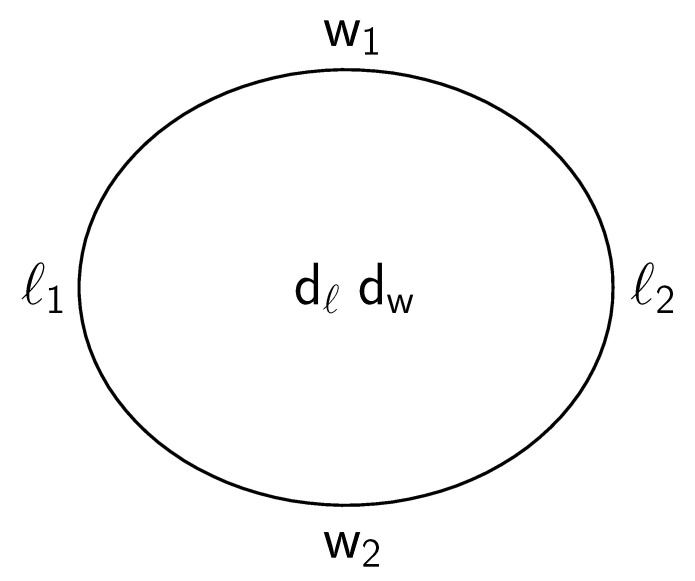
$\ell_{1}$ and $\ell_{2}$ denote the minimal gray values (GVs) along the major axis indicated in Figure 3 (title “long”) by the position of orange lines. By analogy, denote $w_{1}$ and $w_{2}$ the minimal GVs along the minor axis. $d_{\ell }$ and $d_{w}$ denote the maxima in the middle along major and minor axes, respectively. To avoid misunderstanding, minimal GVs indicate maximal RBC thickness.

**Figure 6 cells-11-01941-f006:**
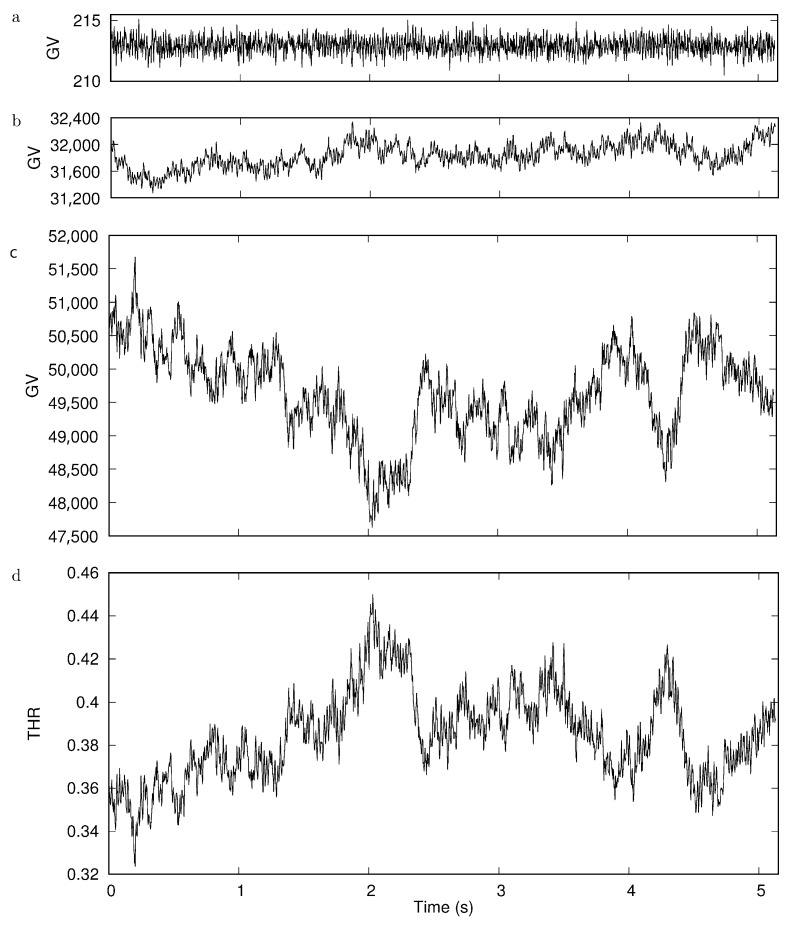
Time series from a movie taken with 400 frames/s. (**a**): Noise of the background. (**b**): Average GV of the four rim positions. (**c**): Average GV of the two dimple positions. (**d**): THR. For details see text.

**Figure 7 cells-11-01941-f007:**
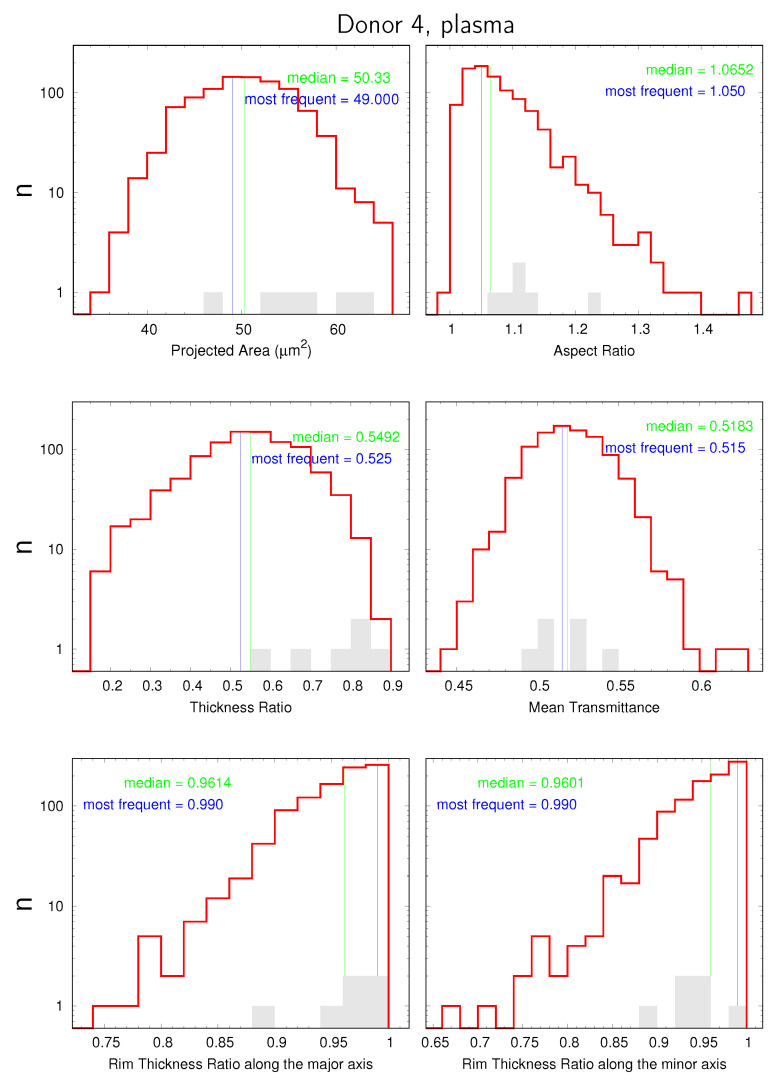
Example of the first page of the data sheets of donor 4. The histogram of normal cells is shown in red. The gray columns indicate the distribution of the E1s found for the respective donor. Blue lines indicate the middle of the most frequent bin. Green lines indicate the exact median value of the respective, red distribution.

**Figure 8 cells-11-01941-f008:**
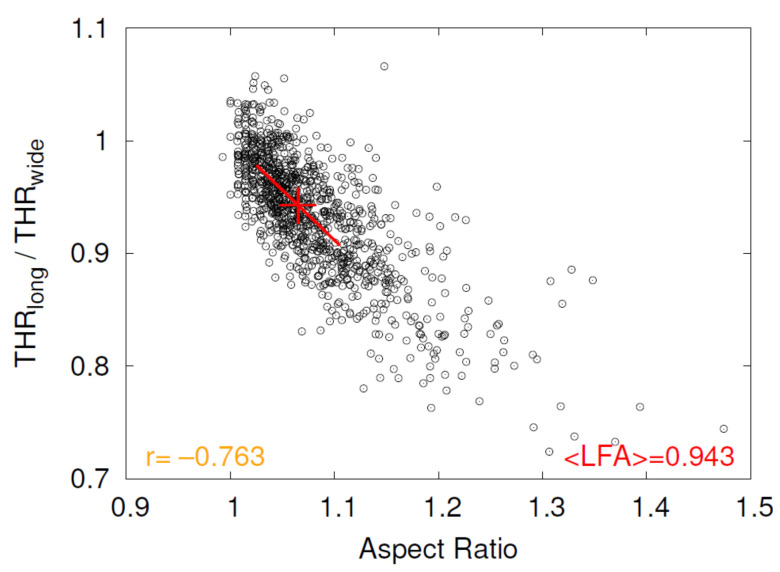
Example of the scatter plot ${\mathrm{THR}}_{\mathrm{long}}/{\mathrm{THR}}_{\mathrm{wide}}$ versus Aspect Ratio of donor 4 for plasma suspension. For details see text.The correlation coefficient is shown in orange in the lower left corner. The red line is a regression of all data with aspect ratios inside the interval 〈median − 0.04, median + 0.04〉, where the median is taken from Figure 7. The red cross indicates the middle of the regression line. The y-position of the red cross denotes the value of LFA corresponding to the median of Aspect Ratio. Its value is shown in red in the lower right corner.

**Figure 9 cells-11-01941-f009:**
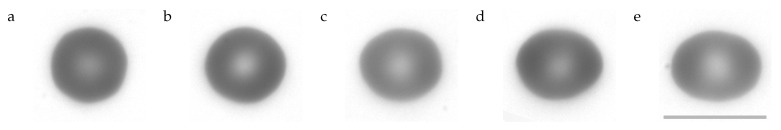
Examples of donor 4 showing the ranges of aspect ratio and <LFA>. Their values are (**a**): 1.0, 1.003. (**b**): 1.073, 0.938. (**c**): 1.146, 0.862. (**d**): 1.219, 0.804. (**e**): 1.291, 0.746. The images are rotated to make the major axis horizontal. The scale bar indicates 10 µm.

**Figure 10 cells-11-01941-f010:**
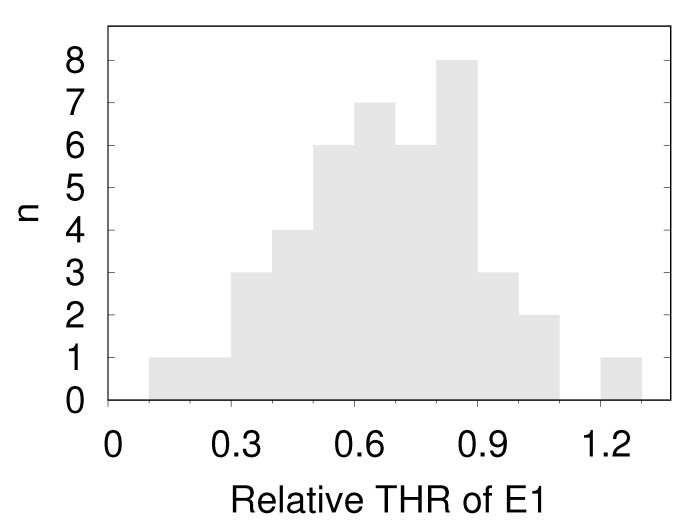
Histogram of the relative position of the THR of the E1s of all donors within the respective interval 〈median of discocytes, largest THR of discocytes〉.

**Figure 11 cells-11-01941-f011:**
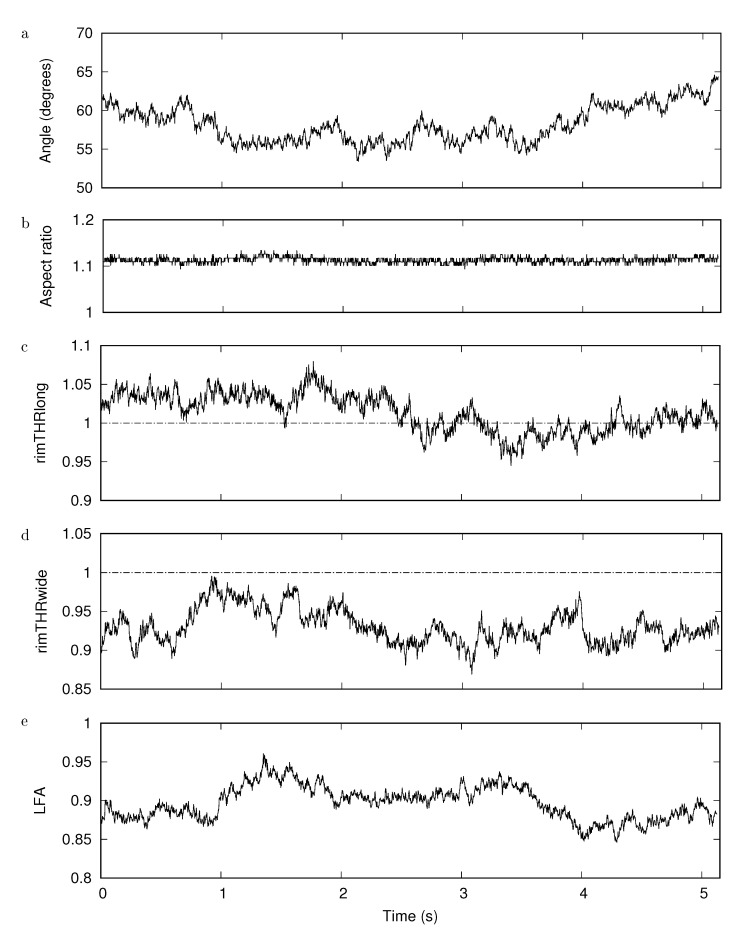
Time series of selected quantities. (**a**): angle of the major axis with the horizontal. (**b**): aspect ratio. (**c**,**d**): Ratio between rim thicknesses at the two opposite ends of the major and minor axis, respectively. In contrast to the definitions in Equations (Equation 7) and (Equation 8), it is here the ratio of two individual thicknesses. (**e**): LFA. For details see text.

**Table 1 cells-11-01941-t001:** Mean values (MV), standard deviations (SD), and coefficients of variation (CV = MV/SD) of the medians of the distributions 1 to 6. Results for RBCs suspended in plasma and serum.

	Plasma, *n* = 8			Serum, *n* = 7		
	MV	SD	CV	MV	SD	CV
projected area	50.97	1.80599	0.03543	50.27	2.63877	0.05249
aspect ratio	1.063	0.00841	0.00791	1.066	0.00882	0.00827
thickness ratio (THR)	0.554	0.06405	0.11561	0.609	0.05487	0.09010
mean transmittance	0.529	0.01832	0.03463	0.527	0.01733	0.03288
rim thickness ratio along the major axis	0.959	0.00644	0.00672	0.960	0.00554	0.00577
rim thickness ratio along the minor axis	0.959	0.00443	0.00462	0.959	0.00521	0.00543

**Table 2 cells-11-01941-t002:** Mean values of the means (MV), mean values of the standard deviations (SD), and mean values of the coefficients of variation (CV = MV/SD) of the distributions 1 to 6. Results for RBCs suspended in plasma and serum.

	Plasma, *n* = 8			Serum, *n* = 7		
	MV	SD	CV	MV	SD	CV
projected area	51.15	5.3243	0.1043	50.45	5.2811	0.1049
aspect ratio	1.077	0.0576	0.0534	1.081	0.0628	0.0580
thickness ratio (THR)	0.545	0.1341	0.2501	0.593	0.1346	0.2294
mean transmittance	0.530	0.0234	0.0440	0.528	0.0257	0.0487
rim thickness ratio along the major axis	0.950	0.0396	0.0418	0.951	0.0393	0.0413
rim thickness ratio along the minor axis	0.950	0.0413	0.0435	0.950	0.0406	0.0427

**Table 3 cells-11-01941-t003:** Mean value (MV), standard deviation (SD), and coefficient of variation (CV = MV/SD) of <LFA>. Results for RBCs suspended in plasma and serum. $\mathrm{LFA}={\mathrm{THR}}_{\mathrm{long}}/{\mathrm{THR}}_{\mathrm{wide}}$.

	Plasma, *n* = 8			Serum, *n* = 7		
	MV	SD	CV	MV	SD	CV
<LFA> = LFA at the median of THR	0.942	0.00845	0.00897	0.944	0.00684	0.00725

**Table 4 cells-11-01941-t004:** Measurements of RBC diameter. The buffered salt solutions used in different experiments are subsumed under saline.

Suspending Medium	Total Number of RBCs	Number of Donors or Groups	RBC Orientation	Optical Method	Mean Value (µm)	SD (µm)	Reference
plasma	1917	6 donors	face-on	bright field image	8.55	0.54	[9]
serum	≈2000	5 donors	edge-on	bright field image	8.55	0.41	[7]
saline	1016	7 donors	edge-on	bright field image	8.069	0.547	[13]
saline	1267	7 donors	face-on	bright field image	8.063	0.429	[13]
saline	50	1 donor	face-on	interference holography	7.82	0.62	[1]
saline	2853	4 groups	face-on	interference holography	7.66	0.67	[2]
saline	>22,000	22 donors	any	light scattering	6.42	0.763	[12]
plasma	4882	8 donors	face-on	distribution of transmittance	8.123	0.425	this work
serum	5029	7 donors	face-on	distribution of transmittance	8.071	0.425	this work

**Table 5 cells-11-01941-t005:** Measurements of the THR.

Suspending Medium	Total Number of RBCs	Number of Donors or Groups	RBC Orientation	Optical Method	Mean Value	Reference
serum	≈2000	5 donors	edge-on	bright field image	0.424	[7]
saline	50	1 donor	face-on	interference holography	0.314	[1]
saline	2853	4 groups	face-on	interference holography	0.509	[2]
saline	25	not specified	face-on	defocusing microscopy	0.50 ^1^	[10]
saline	20	not specified	face-on	diffraction phase microscopy	0.27 ^1^	[11]
saline	>22,000	22 donors	any	light scattering	0.627	[12]
plasma	4882	8 donors	face-on	distribution of transmittance	0.550	this work
serum	5029	7 donors	face-on	distribution of transmittance	0.601	this work

^1^ Measured from the published cross sectional drawing of an average cell.

## Data Availability

Not applicable.

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
