# Peer review of "The Shape of Human Red Blood Cells Suspended in Autologous Plasma and Serum"

_cells, 2022, doi:10.3390/cells11121941_

Round 1

Reviewer 1 Report

I’ve read with great interest the manuscript. There are several issues that prevent acceptance in present form. The main concern is lack of clarity: continuous reference to detail and numbers and formulae in the supplementary material must be avoided. Just give the essential info to understand what measurement was performed and how, and give the final interpretation of these results.

Clarity is also very low in general, starting from the introduction, .The work presented here was aimed at reevaluating from a quantitative point of view, some parameters related with the shape of RBCs, in resting conditions and in different media, from PBS to plasma to serum.

Being a quantitative study, one would expect already from the introduction to see some numbers. If the authors challenges the previous values of thickness ratio (THR), i.e. the ratio of the RBC thickness measured un the dimple to the thickness measured at the rim, then in the introduction it must be said that the average ratio, or the range, in previous literature was 0.xxx whereas in the present work it was 0.yyy or 0.zzz whether measured in plasma or serum.

In the introductions reference to a movie in the supplementary material is made to show that the presence of albumin has an effect on the THR values, in a concentration dependent manner, but, again, no concentration values of BSA (from, to?) are given making it difficult to compare to physiological.

Much confusion from the introduction as to how to interpret this THR. It is said that the literature has a gold standard THR value (what is it?) but this value, from Fung et al., it is said to be “too low”. This term is not very rigorous. Too low with respect to what? Adding to the confusion: the THR appears to be “too low” both in the absence of albumin, and in the presence of “high” concentrations of albumin, and to be “high” (the THR) at intermediate albumin concentrations. What is: “low”, “too low”, “high”, “low” and “intermediate concentration”?

Par 2.3: Image processing “per se”. what does it mean? In this paragraph the image processing shown in figure2 is described, but it should be repeated in the legend to figure 2 to make it clear that Figure 2b IS ROTATED with respect to figure 2a!

In the same paragraph it must be stated clearly that the grey values have to be taken as TRASMITTANCE values, otherwise the clarity of exposure is again compromised (one tends to associate large values of grey with larger absorbance, whereas in this work it is the opposite, making it reading the RBC profiles (see fig 3) quite counterintuitive.

Along this line: in legend to Figure 5 l1 and l2 denote the minimal grey values. Please add: indicating the maximal thickness.

One has to get to paragraph 2.5 to understand that transmittance is used, not absorbance. Please anticipate this concept at an earlier stage.

Paragraph 3.1 of Results. It is said “The distribution of aspect ratios extended in some cases to values < 1 (cf. Figure 6). The origin of this illogicality is due to image processing. The orientation of the major axis is determined from an ellipse fitted to the contour whereas the lengths of the axes and the aspect ratios are determined from the dimensions of a rectangle enclosing the binarized contour.” Please clarify, because both methods should give roughly the same results: minor and major axis of the ellipse should correspond to minor and major sides of the rectangle, respectively.

Table 1 cannot be understood: thickness ratio (THR) is given (line 3), which should correspond to the ratio of the thickness across the dimple to the thickness across the rim. The problem is then that the rim has different thickness along the major and the minor axes. So what is the axis considered for the THR value of table line 3).

In lines 5 and 6 of the table two other values are given, which should be exactly the THR values computed considering the same thickness across the dimple but the two different values of thickness across the major axis rim or minor axis rim. But these values are called “rim thickness ratio along the major axis” and “rim thickness ratio along the major axis”. Why are not they called “THR along major axis” and “THR along minor axis”? If this is the case?

Another discrepancy is evident: why are the two above mentioned values, THR across minor and THR across major axis exactly the same? Shouldn’t they be different as anticipated by the author in 2.5 and figure 5?

The very definition of THRlong and THRwide is very confusing. It is defined as the THR along the major or minor axis. The term “along” leads to believe that it is the thickness that the rim has in the region that goes from two opposite points along the equator of the ellipse. Conversely, the THR wide, defined as the THR along the minor axis, it is the thickness of the rim that runs from the two opposite poles of the ellipse.

However, from figure 5 one must conclude that the actual definition is the exact opposite.

Adding to the confusion is the fact that THR is always a ration between the thickness at the dimple (numerator) and the thickness of the rim. So when one speaks of a THR value larger than another it means, counterintuitively, that the larger the THR the smaller the thickness.

Please clarify all this and, consistently, use appropriate definitions in table 1, and explain why mean values in lines 5 and 6 are the same.

Moreover, data in table 1 must be elaborated statistically to show if there are any statistically significant differences between plasma and serum values.

Table 2 can be easily omitted and only mentioned in the text according to the results of statistical elaboration.

Table 4. Again statistical elaboration is completely missing. Moreover, it is even impossible to evaluate the variability of RBC diameters because only the mean values are given. At least the standard deviation for those values determined by the author must be given.

The same goes for Table 5 with the THR values.

Several unclear points are also in the discussion. Manu undefined terms or physical concepts. For instance at paragraph 4.2, bottom of page 11 “the spontaneous curvature of the RBC membrane is greater in serum compared to the value in plasma”. What is it? It hasn’t been defined before in the MS. And there are two kinds of curvatures, at the rim and in the dimple, with opposite directions. An “increase in the spontaneous curvature” may mean an increase in the absolute value of both, positive and negative curvature regions. Conversely, a decrease in the spontaneous curvature should make the cell more flat. But if a cell becomes more flat, the relative differences in the thickness at the dimple and at the rim should become smaller, with thickness at the dimple increasing and vice versa for the thickness at the rim. The result should be an INCREASE in THR for this particular cell.

However, in the same 4.2 paragraph, one reads “A decrease in spontaneous curvature decreases the THR”, which is the exact opposite of the conclusion reached above. Which is true?

At the end of paragraph 4.3, Correlations, another undefined concept is presented, which requires clarification: “reduced volume”.

The concepts of positive and negative curvature should be defined earlier in the MS if they must be used to make comments later such as in paragraph 4.4.2 in the sentence: “These shapes are interpreted as intermediates of the return to an outline with positive curvature.”.

Paragraph 4.5.1 Another superfluous and overdetailed description of an experiment with results in several supplementary figures. It is almost impossible to follow the arguments in the main text if 2 supplementary figures with a total of 4 subpanels are continually referred to in the text. Please reduce this part to a summary of the aim of the experiment (distinguish between noise of the camera or flickering of the RBC as the cause of variations in grey values), method to asses it and overall conclusion.

Paragraph 4.5.3. Effect of gravity. The conclusion that gravity effect of RBC shape is based on a theoretical result of a study on (artificial?) phospholipid vesicles. But the difference in density between plasma and RBC cytoplasm is much bigger than from plasma and a phospholipid vesicle. The impact of gravity on shape may be significant for RBCs.

Minor

Materials and Methods 2.2 Microscopic observation: it is “silicone grease” not “silicon grease”

At the end of 2.2: “stored in the hard disk of a computer”

Par 2.3: Image processing “per se”. what does it mean?

Image processing was done with “Imagej”.

Pag 14 line 12 of text: “and is in keeping the hypothesis” - - > “and is in keeping with the hypothesis”

Pag 14 line 17 of text: “from the main the cell body” - - > “from the main cell body”.

Author Response

a PDF was submitted

Reviewer 2 Report

The manuscript deals with experimental measurements of the shape of a red blood cell in equilibrium. The key point is that this manuscript in contrast to earlier works uses natural plasma as surrounding medium. The manuscript is well written and carefully carried out. I can warmly recommend publication and only have a few minor suggestions

·      Can the author provide an approximate equation in the style of the Evans equation for the RBC shapes that he finds? If this was possible, I am sure it would make the paper even more useful for the modeling community.

·      Again with an eye on usability, it would be quite helpful to include the raw data (which now is mostly in figures in SI). This could be done as plain text tables, Excel or any other standard format and the files uploaded as SI.

·      Can data from Ref 1 also be included in table 4?

·      Section 4.5.1 is a bit difficult to read as most of the relevant figures are in SI. I suggest to move the figures into the main text.

Author Response

. To give an equation in the style of Evans and Fung for an RBC deviating from circular symmetry is beyond my capabilities. Instead I suggested a series of steps how it could be done with a discrete formulation of the membrane. New text is in Conclusions (green) and in the Supplemental Materials section 12.

. Several zip files are added to the Supplemental Materials and a line (green) added to the section Supplementary Materials.

. Data from Ref 1 are included in Table 4, however, I could not colorize this line inside the table.

. Two figures have been moved to the main text and some text (green) has been added.

Blue and red text is due to the response to reviewers 2 and 3, respectively.
Orange text is due to the requirements of the journal.
Additions of myself are blueGreen.

Reviewer 3 Report

The presented manuscript contains exciting material.
However, to be able to publish it, the author must make several adjustments to the text.

1. It is necessary to ensure that the data given in all tables and graphs are described. For example, the content of Tables 4 and 5 is hardly discussed.

2. It is necessary to dedicate a special section, concentrating the entire discussion on the influence of the composition of the external environment on the shape of the red cell.

3. It is necessary to add a special paragraph dedicated to the approach's limitations.

Author Response

1. I added some text relevant for Tables 1--3 (blue in section 3.3). The new text to Table 4 is in section 4.2 (blue). I added a few lines of text for Table 5 (first § of section 4.4, blue). The second § is also devoted to Table 5 but is not new.

2. I am not sure what is expected with this point. A general discussion of RBC shape change could easily fill a separate paper. Therefore I restricted myself to describe the so-called glass effect (section 4.3). I am willing to write more but need more specific requirements.   

3. I added two sentences in section 2.5 (blue) and the complete section S3 to show that despite the absorption of plasma equation 2 is still valid. I added section 4.1 that describes the limitations of the method.

Green and red text is due to the response to reviewers 1 and 3, respectively.
Orange text is due to the requirements of the journal.
Additions of myself are blueGreen.

Round 2

Reviewer 1 Report

I apologize for having at times misunderstood the manuscript. However, maybe this is because something was not completely clear in the first place. I proposed clarifications that other might find useful.

 Still a few issues:

THR must obviously be dimensionless, also in the abstract.

I’m perfectly aware that the definitions given through equations are perfectly correct and consistent. What was sometimes unclear (and in part still is) is their translation in plain language. For instance, I still find the definition “Rim Thickness Ratio along the major axis = l1/l2 ,when l1 < l2 and vice versa.”  (equation 7, and also 8) confusing.

 I would change it into any of these, or even a better one, definitions:

  “Ratio between rim thicknesses at the two opposite ends of the major axis”

 “Ratio between rim thicknesses at the two extremities of the major axis”

“Ratio between rim thicknesses at the intersection of major axis’ opposite ends with the rim”

 The same goes for the minor axis.

 Moreover, the “vice versa” is clearly superfluous. It is sufficient to say =l1/l2 not “when”, but where l1<l2.

“The spontaneous curvature can be considered as the curvature a piece of membrane would assume upon conceptually excised from the membrane”

-- > “The spontaneous curvature can be considered as the curvature a piece of membrane would assume when conceptually excised from the membrane”
